# Predictors of Clinically Meaningful Results of Bracing in a Large Cohort of Adolescents with Idiopathic Scoliosis Reaching the End of Conservative Treatment

**DOI:** 10.3390/children10040719

**Published:** 2023-04-13

**Authors:** Sabrina Donzelli, Giulia Fregna, Fabio Zaina, Giulia Livetti, Maria Chiara Reitano, Stefano Negrini

**Affiliations:** 1ISICO (Italian Scientific Spine Institute), 20141 Milan, Italy; 2Doctoral Program in Translational Neurosciences and Neurotechnologies, University of Ferrara, 44121 Ferrara, Italy; 3IRCCS Eugenio Medea-Associazione La Nostra Famiglia, 23842 Bosisio Parini, Italy; 4Department of Biomedical, Surgical and Dental Sciences, University “La Statale”, 20122 Milan, Italy; 5IRCCS Istituto Ortopedico Galeazzi, 20157 Milan, Italy

**Keywords:** adolescent idiopathic scoliosis, brace, outcome predictors

## Abstract

Background: We need good outcome predictors to maximize the treatment efficiency of adolescents with idiopathic scoliosis (AIS). The in-brace correction has shown an important predictive effect on brace failure, while the influence of other variables is still debated. We aimed to identify new outcome predictors from a big prospective database of AIS. Methods: Design: Retrospective analysis of prospectively collected data. Inclusion criteria: AIS between 21 and 45°, Risser 0–2, brace prescription during the observation, treatment conclusion. All of the participants followed a personalized conservative approach according to the SOSORT Guidelines. Outcomes: End of growth below 30°–40°–50°. The regression model included age, BMI, Cobb angle, ATR, TRACE score, real brace wear (RBW), and in-brace correction (IBC). Results: A total of 1050 patients, 84% females, ages 12.1 ± 1.1, 28.2 ± 7.9° Cobb. IBC increased by 30%, 24%, and 23% the odds of ending treatment below 30°, 40°, and 50°, respectively. The OR did not change after the covariate adjustment. Cobb angle and ATR at the start also showed a predictive effect. Conclusions: The systematic evaluation of IBC in clinics is useful for individuating the patient response to brace treatment more accurately, even in relation to the Cobb angle and ATR degrees at the start. Further studies are needed to increase the knowledge on predictors of AIS treatment results.

## 1. Introduction

Adolescent idiopathic scoliosis (AIS) is a complex condition that affects 0.47 to 5.2% of the young population [1]. AIS treatment is complex, mostly because the individual risk of progression and aggressiveness is only partially predictable [2,3].

Given the importance of individual variables in affecting the course of the condition, a personalized approach has been proposed by the International Society on Scoliosis Orthopedic and Rehabilitation Treatment (SOSORT) Guidelines [4]. They suggested a step-by-step strategy based on the progression risk of the subject, currently mainly based on the Cobb angle and bone maturity: the higher the risk, the higher the treatment intensity [4]. Bracing is the most important conservative treatment for AIS of medium degree [4,5]. Brace efficacy for controlling AIS progression and reducing the risk of surgery has been recently affirmed [6,7,8]. Some studies have focused on predictors of bracing results other than Cobb angle and bone maturity.

Xu [9] and Van Den Bogaart [10] demonstrated the fundamental role of in-brace correction, even if the minimal rate of correction needed to achieve final good results is still debated [10]. Wong et al. showed that in-brace correction offers relevant information about structural trunk plasticity [11]. Thus, a greater curve flexibility can lead to a better brace correction [12]. However, other spinal and pelvic variables beyond flexibility seem to influence the rate of in-brace correction [12,13], but their predictive role is not well-established. A recent systematic review clarified the predictive value of various radiographical variables on brace outcome in AIS subjects, where reduced in-brace correction and <28% supine flexibility were confirmed to be predictors of progression with high and moderate evidence, respectively [11]. However, the wide heterogeneity between samples in terms of clinical characteristics and brace management influenced the robustness of the results.

Dosage and compliance also play a major role. Weinstein, in “Bracing in Adolescent Idiopathic Scoliosis Trial” [6] (BrAIST study), proved that the treatment success increased with longer hours of brace wear through a significant positive association. Recently, another study [14] compared the results of the BrAIST cohort with a matched cohort from ISICO, Italy. The results demonstrated a significant reduction in the treatment failure rate in the Italian cohort, from 39% of curves above 40° at the end of the brace treatment in BrAIST to 12%. This result was primarily associated with the greater number of bracing hours in Italian patients. Concerning the dosage, it is necessary to also consider the overtreatment risk. Sanders [15] raised the “brace overuse” issue in patients who did not benefit from the brace because their curves would have progressed to a surgical range. A clinical prediction rule was developed and validated for the patients recruited during the BrAIST study refusing brace treatment [16]. The thoracic curve localization, the Cobb angle at the start, and the simplified skeletal maturity scoring system [17] were good predictors of the surgical risks. Karol [18] underlined the association between a high level of compliance and the therapy’s success. As highlighted by Seen, brace compliance represents a valuable predictor in long-term outcomes in terms of the patients’ curve progression and need for surgery [19]. In this regard, clinicians could actively elicit treatment compliance in patients through monitoring during the therapy. Some studies on using iButton in everyday clinics confirmed the positive impact of compliance monitoring [20]: an objective measure of brace wearing through an electronic device allowed clinicians to practice more accurate prescription management. Using iButton also increased the patient’s compliance without affecting the patient–parent or patient–physician relationships. However, in these studies, the results were not discussed in relation to their consequences on brace modality (hours per day).

Finally, no conclusive evidence has been currently achieved on the predictive role of other radiographic variables such as torsion [21], wedging [22,23], intervertebral rotation at the upper and lower junctions of the curve [21], and pelvic and sagittal parameters (pelvic incidence [24,25], sacral slope [24,25], thoracic kyphosis [24], lumbar lordosis [24,25,26], T1 and T9 spinopelvic inclination [24,25]).

To maximize the treatment efficiency, good outcome predictors, key factors for better managing the therapy, are required in our clinical practice. Their knowledge could help clinicians identify the best strategy and treatment intensity for achieving therapeutic goals early. Their identification could explain the outcome differences observed in the current literature and help to understand the different responses to treatment in accordance with individual variability.

The aim of this study was to identify new outcome predictors to maximize treatment efficiency (brace dosage) from a big prospective database of idiopathic scoliosis patients with the final objective of improving a patient’s tailored therapy and avoiding under and overtreatment. In particular, this study aimed to investigate the influence of clinical and radiographic variables on the Cobb degrees achieved at the end of the brace treatment, especially if the thresholds of 30, 40, or 50° Cobb had been reached or not, better investigating their consequences on the curve progression.

## 2. Materials and Methods

### 2.1. Study Design

This was a prognostic study with historical cohort analysis. Data came from a prospective collection of electronic health records, and a systematic collection was undertaken during routine clinical practice, which started in 2003.

### 2.2. Ethics

Informed written consent was obtained from all of the participants, and the study protocol was submitted and approved by the local Ethical Committee before starting the data extraction (authorization code: 801_2015bis).

### 2.3. Setting

The setting was at a tertiary referral institute specialized in conservative scoliosis treatment in Italy. The data extraction and analysis were performed in December 2017.

### 2.4. Participants

Among all of the patients’ records available, we considered all consecutive patients respecting the following inclusion criteria:Diagnosis of adolescent idiopathic scoliosis;Presence of curves between 21 and 45° at the first consultation at our Institute;Risser stage from 0 to 2;Brace prescription during the observation;Treatment conclusion defined as achievement of Risser 3 and/or medical prescription due to no more risks of progression.

We excluded all subjects with brace treatment before the first consultation at our Institute. In the case of missing X-rays due to some mistakes made during data collection, we excluded patients without the X-rays within the three months before/after the start, patients who did not provide all of their clinical and radiographical data since the first consult to the end of treatment, or patients who refused to take the in-brace X-rays at one month of wearing.

### 2.5. Intervention

All of the participants followed a personalized conservative approach according to what was recommended by the SOSORT International Guidelines [4], where treatment intensity increases with the risk of progression.

The braces prescribed were as follows: for curves between 20 and 30 Cobb degrees, the SpineCor elastic brace [27] and Sibilla plastic brace [28] were used, while the Sforzesco brace [28] has been prescribed for curves above 30°. However, for every patient, the specific brace type and dosage (always above 18 h per day at the first prescription) were evaluated in relation to risk factors such as the Risser stage, curve type, curve magnitude, hump magnitude, hump stiffness, and pubertal signs. The therapeutic goals were always shared with the family and the patients, and the decision was made together to balance between the maximum efficacy and minimum burden for the patients. The treatment goals included short and long-term risks, as highlighted by the SOSORT Guidelines [4].

The brace is prescribed full-time at the start for all patients, meaning that the brace dosage is between 20 and 23 h per day. The dosage is then reduced at follow-up, according to the obtained results and the estimated risks, by a maximum of 2 h at each visit. Compliance was kept high throughout the entire treatment path [20]. The follow-up visits were conducted every six months, while the radiographical checks took place every 12 months to reduce the radiation exposure. Exceptions include significant variation in clinical measures, making the medical doctor suspect a sudden progression.

In 2010, we introduced iButton into our clinical practice, and data on real brace wear have become available. In Italy, the temperature sensor has a cost for the patient, while the public health system provides the brace. The cost is low but sometimes not affordable by the patients and their families because they have to cover the costs of the treatment: travelling for physiotherapy, visits far from home, and having access to private health care. These are among some of the reasons why some patients refused to buy a temperature sensor; therefore, only part of the cohort had this data available. However, Donzelli (2012) demonstrated good reliability from the referred wearing compliance to the real one in our cohort [20].

Except for patients with the SpineCor prescription, all the others also performed exercises according to the SEAS (Scientific Exercise Approach to Scoliosis) principles [29]. The physiotherapy protocol included one session with the patient every month and home-based exercises at least five days per week for 20 min to train the ability to perform an active self-correction.

### 2.6. Outcome

The primary outcome was continuous. We defined the primary outcome as the variation of the Cobb angle over time. All measurements were taken by the physicians during the visit and checked randomly by blinded assessors [14].

Secondary outcomes included achieving curve progression under the threshold of 30° at the end of growth. The secondary outcomes were identified by a progression under 40° Cobb and under the surgical threshold of 50° [30].

### 2.7. Covariates

To investigate the factors involved in the therapy success (predictors of result), we applied a regression model that included the following covariates: coming from the medical history: age, gender, and referred brace wear. The covariates from the clinical evaluation were as follows: height (in cm), weight (in kg), body mass index (BMI) (kg/sqm), angle trunk rotation (ATR), defined as hemithorax volume difference resulting from the vertebrae rotation assessed through the Bunnell scoliometer, aesthetic impact measured with the TRACE (Trunk Aesthetic Clinical Evaluation) score [31], and real brace wear (RBW) through the iButton device. Part of the covariates also came from radiographical data: Cobb angle, in-brace correction (IBC) (through in-brace X-rays taken after one month of brace wear). All data were related to the patient’s first evaluation.

### 2.8. Sample Size

During the data extraction, we included all subjects who consented to the study and respected the inclusion criteria. The first premise is that for reliable predictions, the number of events per variable must be ten at minimum. The strength of predictors plays an important role, and the literature review and previous experience show that in the field of scoliosis, the predictors are not very strong. Therefore, the strategy was to have the maximum size available according to the inclusion criteria and to optimize the available data by carefully managing missing data.

### 2.9. Statistical Methods

We prepared descriptive statistics with the mean and standard deviation for continuous data and proportions for categorical data. We checked missing data and described them to estimate the potential limitations in the analysis. Missing data analysis raised concerns about the reliability of the developed model; for the purpose of comparison, sensitivity analysis provided results after the regression imputation of random missing data. A univariate linear regression model was run with the Cobb angle as a continuous outcome variable to explore the performance of every single predictor. This prepared the choice of the independent variable to be introduced into the multivariate linear regression model. Finally, a backward and forward stepwise regression procedure was applied to test the variables in the multivariate linear regression model.

The secondary analysis included the fitting of multiple logistic regression models. The crude and adjusted coefficient guided the selection of the covariates to be included in the model, together with the backward and forward stepwise procedure.

## 3. Results

We included 1050 subjects in the study, 84% females, with a mean age of 12.1 ± 1.1 years. The mean Cobb angle at the maximum curve was 28.5 ± 8.0 Cobb degrees in females and 27 ± 7.9 Cobb degrees in males. Table 1 reports the other demographic characteristics.

We found that the final result was influenced by the starting Cobb angle, with an increase of 0.88° (CI95% 0.78–0.98) at the end for each Cobb angle at the start (Table 2). We also found that the in-brace correction (IBC) and the hump magnitude expressed as the angle of trunk rotation (ATR) were good predictors of the final Cobb angle: the effect of IBC on the final Cobb angle, adjusted for Cobb and ATR at the start, was 0.57 (CI95% 0.44–0.70). Real brace wear (RBW) did not result in a significant predictor of the final Cobb angle. Figure 1 shows the scatter plot of the regression line for IBC, ATR, and Cobb angle at the start.

The final regression equation to predict the Cobb angle at the end of treatment is as follows:END Cobb angle = −51.78 + 0.57 ∗ IBC + 0.44 ∗ Start Cobb angle + 0.41 ∗ ATR + 0.003 ∗ Age at start

After regression imputation, IBC was confirmed to be a good predictor, the Cobb angle at the start lost statistical significance, and ATR showed a smaller predictive effect, as shown in Table 3.

We did not include the Risser stage in the model because it was unbalanced in the sample (i.e., most of the included subjects were at Risser 0 or 1). Nevertheless, we tested the multiple binary comparisons (0 vs. 1; 0 vs. 2), but we did not find further information.

When considering specific clinically significant endpoints (i.e., below 30°, 40°, or 50°), the binary logistic regression results again showed the primary role played by IBC. As the IBC increased, the odds of ending treatment below 30, 40, and 50 degrees were 30%, 24%, and 23% higher, respectively. The odds did not change after the adjustment of the covariates (ending below 30 CI 95% 27–33%—pseudo R2 = 0.41; ending below 40 CI95% 28–21%—pseudo R2 = 0.36; ending below 40 CI 95% 29–18%—pseudo R2 = 0.33).

## 4. Discussion

This study investigated the typical predictors of results in patients with AIS undergoing brace treatment [11].

IBC was among the most powerful predictors of the final results in the analyzed cohort. Thus, the present study confirmed that IBC impacts the final outcome, the curve magnitude, and the hump measure at the first consultation. It is also important to highlight the use of IBC by the team involved in the present study: it is not just a simple check, but the physician and the CPO check the IBC to optimize the correction. This happens more frequently in more severe curves rather than in milder ones.

Results similar to ours have already been reported in the literature. Still, our study had some specificities: we focused on a cohort of patients with very high levels of adherence to prescription and started with full-time brace wear [14,20]. The present results are better than those already reported [20]. Brace dosage has been shown to be a higher impact predictor of results when compared to a cohort treated at a lower dosage [14]. Therefore, multiple factors are involved in determining the results at the end of growth, and this reflects the heterogeneity of results, which characterize many studies attempting to investigate the progression of scoliosis [32,33]. Multiple factors explain the heterogeneity of scoliosis, this is why there is still no clarity in the literature on the predictors, and the results are not coherent among the studies; therefore, this paper contributes to a growing body of literature to better understand the role of bracing.

To our knowledge, this was the first study on brace predictor analysis with the largest single cohort of consecutively recruited AIS patients; we were able to check whether different types of braces influenced the results, contrary to what is usually reported.

Various studies have analyzed the initial Cobb angle effect on brace outcome in AIS patients with different findings [22,23,24,25,34,35,36,37,38,39]. Only some authors have found a significant correlation in terms of curve progression [22,23,34,35]. However, statistical limitations on the applied regression models alter the strength of evidence regarding this variable.

In our sample, the BMI, TRACE score, RBW, and age did not show a relevant effect. Many studies [10] pointed out the clinical need to identify elements that underline the individual variability able to justify different outcomes in cohorts of patients with the same characteristics. Recently, IBC has been proven to be a strong predictor of brace treatment failure [11], while the influence of other variables (like age, ATR, and BMI) is still debated [10].

The influence of IBC on the final clinical outcome and its correlated impact on the brace failure previously demonstrated [10] suggests routinely considering it in clinical practice. We found that the IBC predictive effect was independent of the differences in brace materials and the design in our sample. This is typical of the personalized approach proposed by the current Clinical Guidelines [4] used in many centers and that we recently found to be more effective than a standardized approach [40]. This approach requires physicians to tailor brace prescriptions for the dosage and type according to specific clinical characteristics and the evaluated risk factors (e.g., curve rigidity or the presence of mixed pathologies). This approach greatly contributes to the good compliance to brace wear shown in previous studies [18]. The fact that the specialist doctor chooses the best brace, according to their expertise, to achieve the best possible outcome for every patient makes null the differences in type, material, design, and construction among braces. The commonality is the careful choice aimed at optimizing the effect of treatment while reducing the burden to the minimum, which is in agreement with the guidelines for conservative treatment [4].

Overall, the present results show that the severity of spine curvature at the first consultation, represented by the Cobb angle and ATR, provides key information about the estimated treatment success rate. The prediction highlights that the power of these variables is better than the effect of other variables such as age, BMI, and aesthetic impairment at the initial consultation. The curve magnitude, defined by the Cobb angle, and partially linked to ATR, may be intrinsically correlated to the rate of in-brace correction obtained. Therefore, a potential overfitting effect of the model should be considered.

Unexpectedly, the RBW did not show a predictive effect on treatment success, contrary to what was recently affirmed in patients with curve progression above 40° at the end of growth [14]. However, in this sample, a small proportion of subjects had data from the iButton. This occurred because we considered a large sample consecutively collected over many years. Many patients started the treatment before 2010, when the iButton was implemented as a standard of care. In this case, a large amount of missing data could have affected the model’s results.

Furthermore, it is essential to highlight that all the patients wore the brace at a very high dosage. This homogeneity means that the variable related to dosage does not influence the measures of the results.

Clinical experience has taught us that multiple factors are involved at the end of growth results. Some of them are still unknown or need to be further investigated, while others are too heterogeneous to determine a significant impact on the results. For example, heterogeneity characterized the sport and leisure engagement of the included patients. In Italy, sports activities are mainly performed outside of school, and many patients abandon the sport at 15 years of age, before the end of the treatment. This is why the sport factor could not be included in the model. The underrepresentation of Risser 1 and 2 is another reason behind the insignificant effect of the bone maturity score.

### Strengths and Limitations of the Study

Concerning the study weaknesses, the high rate of missing data across the entire sample has to be considered an important limitation; even after a regression imputation model was used to solve the missing information, the interpretation of results requires caution. The sensitivity analysis showed no differences in the model run after imputation. However, regression imputation of a large proportion of missing data may have introduced a significant bias in the results.

A possible bias of the study is the absence of blind assessors for the X-ray measurements, which is not applicable in a clinical setting where physicians directly manage the patient’s therapy, even if past tests conducted at our center demonstrated good inter-rater reliability [14].

Another potential bias relates to the high specialization of the rehabilitation center where the patients had been recruited. This introduces selection bias, as most patients come to the Institute after multiple consultations. In fact, most cases can be considered to be among the more severe forms of scoliosis.

Among the strengths of the current study, the data collection from the everyday clinical practice of consecutive patients allows for the realistic applicability and generalization of results. Moreover, this study investigated the predictive role of clinical and radiographic variables in a large cohort of patients consistently conservatively treated through a personalized evidence-based approach [40]. Moreover, as far as we know, this is the study with the largest single cohort of AIS patients in brace predictor analysis.

## 5. Conclusions

The present results confirm the relevant role of IBC in predicting the end of growth results. In the current cohort, IBC was always used to optimize the brace effects, and most of the time, some adjustments were made by the CPO working in a team with the physician. This happens more frequently in severe curves than in milder ones. We recommend IBC evaluation at the beginning of the treatment to check brace correction, optimize it whenever needed, and to have a clear idea of the potential correction (i.e., flexibility) that the brace can induce. The awareness of the correction also motivates the patient to wear the brace consistently.

Concerning clinical research, further studies are required to increase the knowledge about the predictors of results in people with AIS to increase treatment specificity and personalization. Comparisons between different patient cohorts could be helpful to underline the influence of multiple variables and approaches.

## Figures and Tables

**Figure 1 children-10-00719-f001:**
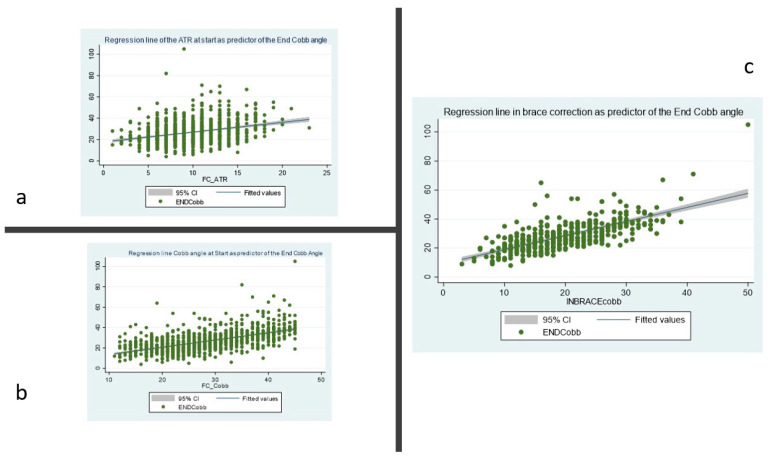
Two-way scatter plot of the regression line for the angle of trunk rotation (ATR) (**a**), Cobb Angle (**b**), and in-brace correction (**c**) at the start.

**Table 1 children-10-00719-t001:** Baseline characteristics of the subjects included.

Characteristics	Females N = 882 (84.0%) Mean (SD)	Males N = 168 (16.0%) Mean (SD)	Entire Sample 1050 Subjects Mean (SD)
Age	12.1 (1.0)	13.0 (1.0)	12.1 (1.1)
Weight	45.9 (8.4)	54.6 (10.6)	47.3 (9.3)
Height	156.3 (2.1)	166.3 (9.7)	157.9 (9.1)
TRACE (Rasch)	6.4 (2.1)	6.1 (2.2)	6.4 (2.1)
Cobb max	28.5 (8.0)	27.0 (7.9)	28.2 (7.9)
ATR	9.7 (3.3)	9.4 (3.6)	9.6 (3.4)

**Table 2 children-10-00719-t002:** The crude and adjusted coefficient of the multivariable linear regression model of the Cobb angle at the end of treatment. RBW = Real Brace Wear, IBC = in-brace correction, in Cobb degrees difference and percentage. * Level of significance set to <0.05. RBW, age at start BMI, and TRACE were automatically dropped by the backward and forward stepwise procedures.

END Cobb Angle	Crude Coefficient (95% CI)	Adjusted R2	*p*-Value	Adjusted Coefficient in Stepwise Forward and Backward Multiple Regression (95% CI)	Adjusted R2	*p*-Value
IBC	0.89 (0.79–0.99)	0.44	<0.001 *	0.57 (0.44–0.70)	0.53	<0.001 *
Start Cobb angle	0.88 (0.78–0.98)	0.42	<0.001 *	0.44 (0.31–0.57)	0.53	<0.001 *
ATR	0.95 (0.68–1.21)	0.12	<0.001 *	0.41 (0.20–0.61)	0.53	<0.001 *
Age at start	−0.003 (−0.005–0.001)	0.03	<0.001 *	−0.003 (−0.005—0.001)	0.53	<0.001 *
RBW	10.02 (2.16–17.88)	0.01	0.013 *	−3.03 (−8.83–2.77)	0.53	0.306
BMI	7.72 (−8.73–24.18)	0.0004	0.356	−11.90 (−23.62—0.11)	0.53	0.05
TRACE	0.47 (0.02–0.92)	0.009	0.042 *	−0.08 (−0.41–0.25)	0.53	0.652

**Table 3 children-10-00719-t003:** The crude and adjusted coefficients of the linear multivariable regression model after the regression imputation procedure. RBW = real brace wear, IBC = in-brace correction, in Cobb degrees difference, and in percentage. * Level of significance set to <0.05.

END Cobb Angle	Crude Coefficient (95% CI)	Adjusted R2	*p*-Value	Adjusted Coefficient in Stepwise Forward and Backward Multiple Regression (95% CI)	Adjusted R2	*p*-Value
IBC	1.22 (1.17–1.28)	0.63	<0.001*	1.14 (1.10–1.20)	0.61	<0.001 *
Start Cobb angle	0.71 (0.65–0.77)	0.33	<0.001 *	−0.02 (−0.10–0.60)	0.61	0.628
ATR	0.92 (0.75–1.09)	0.10	<0.001 *	0.19 (0.07–0.32)	0.61	<0.001 *
Age at start	−0.003 (−0.005–0.001)	0.03	0.002 *	−0.002 (−0.003–0.001)	0.61	<0.001 *
RBW	6.30 (0.98–11.63)	0.004	0.02 *	−4.04 (−7.48–0.61)	0.61	0.021 *
BMI	6.31 (−4.20–16.81)	0.0004	0.239	−11.03 (−17.53–4.54)	0.61	0.001 *
TRACE	0.50 (0.23–0.78)	0.01	<0.001 *	−0.13 (−0.32–0.065)	0.61	0.191

## Data Availability

The data presented in this study are openly available in Zenodo at https://doi.org/10.5281/zenodo.7454708 (accessed in 18 December 2022).

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
