# Peer review of "Predictors of Clinically Meaningful Results of Bracing in a Large Cohort of Adolescents with Idiopathic Scoliosis Reaching the End of Conservative Treatment"

_children, 2023, doi:10.3390/children10040719_

Round 1
Reviewer 1 Report
Dear Authors,
It is my pleasure to review your study but I have a few doubts.
General information:
-the cited literature is poor, it should be supplemented.
-references should be newer, please correct it.
Introduction:
-the introduction is too general, it would be good to give more precise details about AIS,
M&M:
- for better visualization, please divide the M&M section more precisely,
- in "Study design and ethical issues", please add Ethics Committee approval number,
-the inclusion and exclusion criteria are unclear, it should be corrected,
-lin line 121: "This study aims to investigate the influence of these variables on the final outcome, better inquiring their consequences on the curve progression." What does mean "final outcome" ?
Results:
-the presented results are for the whole group, separate calculations for girls and boys should be performed,
-in addition, other variables should be taken into account in the study, e.g. patients' lifestyle, sports activities, etc., which may affect the results.
Discussion:
-the discussion is based on 3 references, it should be absolutely corrected and expanded,
-limitations of the study should be presented more precisely.
Author Response
Predictors of clinically meaningful results of bracing in a cohort of 1050 adolescents with idiopathic scoliosis
Response to #reviewer1
General information:
- The cited literature is poor, it should be supplemented.
We thank Reviewer1 for this comment, useful for improving our work’s quality. We have implemented the reported literature on this topic.
- References should be newer, please correct it
Thanks Reviewer1 for the suggestion, an update of the available literature has been done.
Introduction:
- The introduction is too general, it would be good to give more precise details about AIS
Thanks for having pointed out this need for a better insight on the theme, we have expanded the text.
M&M:
- For better visualization, please divide the M&M section more precisely
Thanks for this suggestion, we followed the Strobe checklist to organize the M&M section.
- In "Study design and ethical issues", please add Ethics Committee approval number
Thanks for the comment, the approval code has been added.
- The inclusion and exclusion criteria are unclear, it should be corrected
Thanks for this note, inclusion and exclusion criteria have been detailed.
- In line 121: "This study aims to investigate the influence of these variables on the final outcome, better inquiring their consequences on the curve progression." What does mean "final outcome"?
We thank Reviewer1 for this clarification request, the aim of the study has been specified further.
Results:
- The presented results are for the whole group, separate calculations for girls and boys should be performed
Thanks for this suggestion, we added in the demographic table the males and females baseline characteristics. We would not go further as gender did not resulted a significant predictors and males have same characteristics at baseline as females.
- In addition, other variables should be taken into account in the study, e.g. patients' lifestyle, sports activities, etc., which may affect the results
Thanks this is a very good point, unfortunately this information was not available for this sample. We added a comment on this on the limitation of the study in the discussion section.
Discussion:
- The discussion is based on 3 references, it should be absolutely corrected and expanded
Thanks to Reviewer1 for this constructive comment, a review of discussion section has been performed.
- Limitations of the study should be presented more precisely
Thanks for the note, study limitations have been implemented.
Reviewer 2 Report
Dear authors, the present paper deals with an interesting topic.
However, the number of self-citations (7 out of 16) observed in the manuscript references is abnormally high!
Please reduce the number of self-citations and cite papers dealing with the conservative treatment of spine deformities written by other national and international research groups, to improve the quality and reliability of your manuscript.
Author Response
Predictors of clinically meaningful results of bracing in a cohort of 1050 adolescents with idiopathic scoliosis
Response to #reviewer2
Dear authors, the present paper deals with an interesting topic.
However, the number of self-citations (7 out of 16) observed in the manuscript references is abnormally high!
Please reduce the number of self-citations and cite papers dealing with the conservative treatment of spine deformities written by other national and international research groups, to improve the quality and reliability of your manuscript.
We thank Reviewer2 for the comment, helpful for a better characterization of the study focus, the literature on that theme has been expanded.
Reviewer 3 Report
1) Authors are suggested to modify the title and remove the count i.e., 1050.
2) The introduction section should mention more literature which have studied these predictors.
3) Authors need to describe the data collection methodology in detail. If some standard questionnaires or templates were used, mention the template in your article.
4) LINE 83 to LINE 89: Authors need to further explain why these inclusion and exclusion criteria were considered. How are these conditions relevant to the masses? To make the predictors useful for the AIS population, the above need to be explained.
5) Under Outcomes section, how did the authors verified that the recorded data by physicians was correct and based on a selected protocol. Under Sample Size section, clearly mention the sample size, and standard error means (along with the standard deviation) amongst the groups i.e., males and females.
6) Authors need to clearly mention the regression model through equations and further explanations.
7) LINE 202 to 207: The mentioned limitations are critical, and the authors need to further explain how the current study is useful despite these limitations.
8) Some minor changes such as appropriate line spaces and indents as per the journal’s guidelines.
Author Response
Predictors of clinically meaningful results of bracing in a cohort of 1050 adolescents with idiopathic scoliosis
Response to #reviewer3
- Authors are suggested to modify the title and remove the count i.e., 1050
Thanks for suggesting this, we have changed the title into: Predictors of clinically meaningful results of bracing in a large cohort of adolescents with idiopathic scoliosis reaching the end of conservative treatment.
- The introduction section should mention more literature which have studied these predictors
Thank Reviewer3 for the suggestion, a deeper description of the literature available on brace predictors in AIS has been performed.
- Authors need to describe the data collection methodology in detail. If some standard questionnaires or templates were used, mention the template in your article
Thank for this comment, probably it was not clear in the method section that data came from prospective collection of electronic health records collected during clinical routine practice which started in 2003.
- LINE 83 to LINE 89: Authors need to further explain why these inclusion and exclusion criteria were considered. How are these conditions relevant to the masses? To make the predictors useful for the AIS population, the above need to be explained
Thanks to the reviewer for the comment, this is certainly a very crucial point for the relevance and generalizability of our results, thus, the inclusion and exclusion criteria have been furtherly described.
- Under Outcomes section, how did the authors verified that the recorded data by physicians was correct and based on a selected protocol.
Thanks for this comment, data came from routine clinical activity. The data are checked systematically to ensure the quality of data.
- Under Sample Size section, clearly mention the sample size, and standard error means (along with the standard deviation) amongst the groups i.e., males and females
Thanks for pointing this out, we reworded the sentence.
- Authors need to clearly mention the regression model through equations and further explanations
Thanks, we added the linear regression equation.
- LINE 202 to 207: The mentioned limitations are critical, and the authors need to further explain how the current study is useful despite these limitations
Thanks for this helpful suggestion, limitations have been implemented.
- Some minor changes such as appropriate line spaces and indents as per the journal’s guidelines
We thank reviewer3 for the note, a technical revision of the text and the layout has been executed.
Round 2
Reviewer 1 Report
Dear Authors,
thank you for the changes made.
The manuscript looks much better.
Best regards
Author Response
Thank you for your suggestions
Reviewer 2 Report
Thank you for allowing me to review the revised version of the manuscript.
The paper deals with a very interesting topic, however, the scientific quality of the paper should be improved before considering it for publication.
1. Results are not clearly presented. Please rewrite the results section and add some figures that summarize better the findings of this study
2. Discussion: please highlight the innovations shown in this paper (if any!) and describe better strengths and limitations of the study
3. The number of self-citation is still abnormally high (8 out of 35; 22.86%), a paper with such a kind of reference is not suitable for publication!
Author Response
Please see the attachment
Response to #reviewer2
The paper deals with a very interesting topic, however, the scientific quality of the paper should be improved before considering it for publication.
Thank you for providing a new review allowing us to improve even more our paper.
- Results are not clearly presented. Please rewrite the results section and add some figures that better summarize the findings of this study.
Thank you. We totally rewrote the results and added Figure 1.
- Discussion: please highlight the innovations shown in this paper (if any!) and describe better strengths and limitations of the study
Thank you for your (kind) remark. It allowed us to better clarify with a specific statement the contribution of our study to the current literature. We also rewrote the discussion to make it clearer. We added a specific paragraph about strengths and limitations.
- The number of self-citation is still abnormally high (8 out of 35; 22.86%), a paper with such a kind of reference is not suitable for publication!
We are sorry, but we could not manage this issue better than what reported below. Surely, the reviewer recognises the limitation of the highly relevant general rule he report. These limitations are particularly important in little fields of research. We performed a “quick and dirty search” in PubMed as follows to try to clarify the issue.
Search strings in PubMed:
- "scoliosis"[Mesh] AND "idiopathic" and "Braces"[Mesh]
- "scoliosis"[Mesh] AND "idiopathic" and "Braces"[Mesh] AND ("Negrini" OR "Donzelli" OR "Zaina")
Since the search involved all scoliosis research fields, including basic science, and we focus only on clinical research, we also added the following filters
- Clinical Study, Clinical Trial, Comparative Study, Consensus Development Conference, Controlled Clinical Trial, Evaluation Study, Letter, Guideline, Meta-Analysis, Multicenter Study, Observational Study, Practice Guideline, Pragmatic Clinical Trial, Randomized Controlled Trial, Review, Systematic Review.
Results
|
SEARCH |
STRING |
SINCE THE START OF PUBMED |
LAST 10 YEARS IN PUBMED |
||
|
|
|
Total |
With clinical filters |
Total |
With clinical filters |
|
TOTAL OF PAPERS |
1 |
868 |
328 |
349 |
120 |
|
PAPERS OF OUR GROUP |
2 |
42 |
24 |
22 |
14 |
|
PERCENTAGE of the total of papers published by our group |
4.8% |
6.5% |
7.3% |
11.7% |
|
This shows that our group has been leading research on bracing for scoliosis since the beginning of this millennium, well before the publication of the Weinstein study in the NEJM that changed the perspective of many researchers, driving them back to this under-searched area.
We also have another external independent source information. According to Expertscape, 3 authors of this paper are in the first 4 positions for the number of publications in this specific area (https://expertscape.com/ex/braces).
Finally, if we consider prediction models, even less groups are involved. In fact, in our references, there are other groups with multiple papers cited: 4 papers by Qiu Y and Cheng JCY group, 3 papers by Cheung JPY group, and 2 papers each by Dolan LA and Weinstein SL group, Karol LA group and Sanders JO group.
Nevertheless, we tried to comply with the reviewers’ requests, and we list below what we did for each cited paper of ours. We found:
- 3 key papers in the sector
- 4 are key papers in the methods section to understand the instruments we used
- 2 papers that we finally and regrettably decided to excluded: this do not allow to describe all the relevant literature on brace sensors, but we hope this can satisfy the reviewer
Here are the citations one by one.
- Negrini, S.; Donzelli, S.; Aulisa, A.G.; Czaprowski, D.; Schreiber, S.; de Mauroy, J.C.; Diers, H.; Grivas, T.B.; Knott, P.; Kotwicki, T.; et al. 2016 SOSORT Guidelines: Orthopaedic and Rehabilitation Treatment of Idiopathic Scoliosis during Growth. Scoliosis Spinal Disord 2018, 13, doi:10.1186/s13013-017-0145-8.
Key paper in the sector (276 citations this edition + 283 the first edition – source Scopus). Current clinical Guidelines on the topic providing all relevant clinical and therapeutic information.
- Negrini, S.; Minozzi, S.; Bettany-Saltikov, J.; Chockalingam, N.; Grivas, T.B.; Kotwicki, T.; Maruyama, T.; Romano, M.; Zaina, F. Braces for Idiopathic Scoliosis in Adolescents. Cochrane Database Syst Rev 2015, CD006850, doi:10.1002/14651858.CD006850.pub3.
Key paper in the sector (41 citations this edition + 70 the first edition + 26 and 73 the short report in Spine Journal, respectively – source Scopus). Cochrane Review on braces – as you know, Cochrane Reviews are the best quality systematic reviews showing the current best possible evidence
- Donzelli, S.; Zaina, F.; Negrini, S. In Defense of Adolescents: They Really Do Use Braces for the Hours Prescribed, If Good Help Is Provided. Results from a Prospective Everyday Clinic Cohort Using Thermobrace. Scoliosis 2012, 7, 12, doi:10.1186/1748-7161-7-12.
Key paper in the material and methods section. It presents the brace sensor used and how it showed to be reliable in our cohort.
- Negrini, S.; Marchini, G.; Tessadri, F. Brace Technology Thematic Series - The Sforzesco and Sibilla Braces, and the SPoRT (Symmetric, Patient Oriented, Rigid, Three-Dimensional, Active) Concept. Scoliosis 2011, 6, 8, doi:10.1186/1748-7161-6-8.
Key paper in the material and methods section. It presents the braces used. Our group has developed them, and now they are used and reported in the literature by others too. Nevertheless, this is the paper presenting them.
- Romano, M.; Negrini, A.; Parzini, S.; Tavernaro, M.; Zaina, F.; Donzelli, S.; Negrini, S. SEAS (Scientific Exercises Approach to Scoliosis): A Modern and Effective Evidence Based Approach to Physiothe,rapic Specific Scoliosis Exercises. Scoliosis 2015, 10, 3, doi:10.1186/s13013-014-0027-2.
Key paper in the material and methods section. It presents the exercises we used. Our group has developed them, and now they are used and reported in the literature by others too. Nevertheless, this is the paper presenting them.
- Negrini, S.; Donzelli, S.; Di Felice, F.; Zaina, F.; Caronni, A. Construct Validity of the Trunk Aesthetic Clinical Evaluation (TRACE) in Young People with Idiopathic Scoliosis. Ann Phys Rehabil Med 2020, 63, 216–221, doi:10.1016/j.rehab.2019.10.008.
Key paper in the material and methods section. It presents the last methodological advancement of the TRACE measurement. Our group has developed it, and it is now used by others too. We should have cited here also the original paper (Zaina et al. Scoliosis 2009), but we did not, even if it would have contributed more to our personal h-indexes.
- Negrini, S.; Donzelli, S.; Negrini, F.; Arienti, C.; Zaina, F.; Peers, K. A Pragmatic Benchmarking Study of an Evidence-Based Personalised Approach in 1938 Adolescents with High-Risk Idiopathic Scoliosis. J Clin Med 2021, 10, 5020, doi:10.3390/jcm10215020.
Key paper in the sector (2 citations – source Scopus). To our knowledge, this is the only paper in the literature presenting the efficacy of a personalised approach (based on shared-decision making) of bracing using an original methodology. We are ready to change this citation if another one could be suggested.
- Donzelli, S.; Zaina, F.; Martinez, G.; Di Felice, F.; Negrini, A.; Negrini, S. Adolescents with Idiopathic Scoliosis and Their Parents Have a Positive Attitude towards the Thermobrace Monitor: Results from a Survey. Scoliosis Spinal Disord 2017, 12, 12, doi:10.1186/s13013-017-0119-x.
- Donzelli, S.; Zaina, F.; Minnella, S.; Lusini, M.; Negrini, S. Consistent and Regular Daily Wearing Improve Bracing Results: A Case-Control Study. Scoliosis Spinal Disord 2018, 13, 16, doi:10.1186/s13013-018-0164-0.
We also cited all the papers in the literature on the use of thermosensor to check compliance. We finally decided to cut 2 papers of ours.

Round 3
Reviewer 2 Report
Thankyou for giving me the possibility to review the revised version of the manuscript. The paper has been improved, but it does not add anything new to the conservative management of scoliosis.
Moreover, the self-citation rate (7 out of 33) is still abnormally high.
Please reduce the number of self-citations to two and resubmit the paper.
Author Response
We performed a wide series of corrections that you can see in the text with the "revision mode". Now the English should be ok. For what our citations are concerned, we already explained that they relate only to 1) essential papers in the field (e.g. Guidelines) or 2) papers explaining the methods and outcome instruments used in this cohort